# Examining the diagnostic accuracy of artificial intelligence for detecting dental caries across a range of imaging modalities: An umbrella review with meta-analysis

Sarah Arzani[1]*, Ali Karimi[2], Pedram Iranmanesh[3], Maryam Yazdi[1,4,5],
Mohammad A. Sabeti[6], Mohammad Hossein Nekoofar[7], Jafar Kolahi[8]*, Heejung Bang[9],
Paul M.H. Dummer[10]

1 Child Growth and Development Research Center, Research Institute for Primordial Prevention of Non-Communicable Disease, Isfahan University of Medical Sciences, Isfahan, Iran, 2 Maxillogram Maxillofacial Surgery, Implantology and Biomaterial Research Foundation, Istanbul, Turkey, 3 Department of Endodontics, Dental Research Center, Dental Research Institute, School of Dentistry, Isfahan University of Medical Sciences, Isfahan, Iran, 4 Diabetes Research Centre, College of Life Sciences, University of Leicester, Leicester, United Kingdom, 5 Leicester Diabetes Centre, Leicester General Hospital, University Hospitals of Leicester NHS Trust, Leicester, United Kingdom, 6 Advanced Specialty Program in Endodontics, UCSF School of Dentistry, University of California, San Francisco, California, United States of America, 7 Department of Endodontics, School of Dentistry, Tehran University of Medical Sciences, Tehran, Iran, 8 Independent Research Scientist, Founder of Dental Hypotheses, Isfahan, Iran, 9 Division of Biostatistics, Department of Public Health Sciences, School of Medicine, University of California, Davis, California, United States of America, 10 Emeritus Professor of Restorative Dentistry, School of Dentistry, College of Biomedical and Life Sciences, Cardiff University, Cardiff, United Kingdom

* sa.arzan@yahoo.com (SA); kolahi_jafar@yahoo.com (JK)

## Abstract

The objective of this systematic review was to systematically collect and analyze multiple published systematic reviews to address the following research question "Are artificial intelligence (AI) algorithms effective for the detection of dental caries?". A systematic search of five electronic databases, including the Cochrane Library, Embase, PubMed, Scopus, and Web of Science, was conducted until October 15, 2024, with a language restriction to English. All fourteen systematic reviews which assessed the performance of AI algorithms for the detection of dental caries were included. From 137 primary original research studies within the systematic reviews, only 20 reported the data necessary for inclusion in the meta-analysis. Pooled sensitivity was 0.85 (95% Confidence Interval (CI): 0.83 to 0.93), specificity was 0.90 (95% CI: 0.85 to 0.95), and log diagnostic odds ratio was 4.37 (95% CI: 3.16 to 6.27). Area under the summary ROC curve was 0.86. Positive post-test probability was 79% and negative post-test probability was 6%. In conclusion, this meta-analysis has revealed that caries diagnosis using AI is accurate and its use in clinical practice is justified. Future studies should focus on specific subpopulations, depth of caries, and real-world performance validation to further improve the accuracy of AI in caries diagnosis.

**Data availability statement:** All relevant data are within the manuscript and its Supporting Information files.

**Funding:** The author(s) received no specific funding for this work.

**Competing interests:** The authors have declared that no competing interests exist.

## Introduction

On a global level, it is estimated that dental caries affects the permanent dentition of approximately 2.3 billion adults and the primary dentition of approximately 530 million children [1]. The annual treatment costs of dental caries for individuals aged 12–65 years worldwide is estimated at US$357 billion (331 billion), or 4.9% of global health-care expenditure [2].

Early and precise detection of dental caries can lead to effective prevention and treatment with less invasive methods, potentially resulting in improved outcomes and reduced healthcare costs [3]. The conventional strategy for diagnosis of carious lesions is visual examination, supplemented with intraoral radiographs, preferably bitewings [4]. Obviously, the level of clinical expertise of dentists may affect the reliability and accuracy of visual examination methods. The meta-analysis of Walsh T, et al., revealed a pooled sensitivity of visual caries diagnosis of 0.83 and specificity of 0.81 [5]. The meta-analysis of Schwendicke F, et al., assessed diagnostic accuracy of intraoral radiography, i.e., bitewing or periapical radiographs. They reported the pooled sensitivity for radio-graphic detection of any type of occlusal carious lesion in clinical studies was 0.35, and pooled specificity was 0.78. For radiographic detection of any type of proximal caries in clinical studies, the pooled sensitivity was 0.24, and pooled specificity was 0.97 [6].

Development of reliable, automated, user-friendly, and low-cost tools for diagnosis of dental caries can play an important role in the management of the disease, and improve oral healthcare access and quality globally. Artificial intelligence (AI) algorithms, particu-larly convolutional neural networks (CNN), are revolutionizing dental care [7]. AI algo-rithms can detect dental caries from various imaging modalities, such as intraoral photographic images, periapical radiographs, bitewing radiographs, panoramic radio-graphs, cone beam computed tomography (CBCT) and near-infrared-light transillumi-nation. As an example, the "Videa Dental Assist" is an AI-based caries detection system approved by the U.S. Food and Drug Administration (FDA) that can analyze bitewing, periapical, and panoramic radiographs acquired from patients aged 3 years or older [8].

AI algorithms can automate the diagnostic process, reducing reliance on human expertise, and be used on smart phone apps [9] or cloud platforms, enhancing access to dental care particularly in underserved regions. Despite the promising results, challenges remain in diagnostic accuracy and generalizability of AI platforms across diverse populations and imaging modalities.

Several systematic reviews have attempted to answer concerns that surround the accuracy of AI algorithms as a diagnostic tool for caries detection. The systematic reviews rarely undertook meta-analyses and pooled sensitivity and specificity values were therefore unavailable for evidence-based clinical decision making. The reviews looked at a range of factors when using AI for detection of dental caries, which has led to an overlap of the primary studies in several reviews. Therefore, the aim of the present study is to systematically collect and assess multiple published systematic reviews to answer the question "Are AI algorithms effective for the detection of dental caries?" by including a meta-analysis and reporting pooled sensitivity, specificity and diagnostic odds ratio.

## Materials and methods

### Protocol registration

The protocol of the review was registered in PROSPERO (#CRD42024568618). The Preferred Reporting Items for Overviews of Reviews (PRIOR), the Preferred Reporting Items for Systematic reviews and Meta-Analyses of Diagnostic Test Accuracy Studies (PRISMA-DTA) and PRISMA-AI criteria were followed to provide accurate and transparent reporting of the review methodology and results [10–12].

### Eligibility criteria

The selection of studies on the accuracy of diagnostics are based on the PIRD criteria, which include the population, index test, reference test, and diagnosis of interest [13]. The utilization of the PIRD format is suggested as a framework for defining the inclusion criteria used in systematic reviews focusing on the accuracy of diagnostic tests [13]. The PIRD elements of the research were defined as: Population: Systematic reviews and meta-analyses evaluating the diagnostic accuracy of AI algorithms for dental caries detection, Index Test: Application of AI algorithms in dental caries detection, Reference Test: Visual-tactile or clinical based assessment of dental caries, Diagnosis of Interest: The diagnostic accuracy measures of the AI model for dental caries including sensitivity, specificity, log diagnostic odds ratio, and area under receiver operating characteristic (ROC) curve (AUC).

All systematic reviews involving human subjects and relevant dental images, reporting performance metrics or results of AI algorithms in dental caries detection were included. Research that covered non-AI methods for detecting dental caries were excluded. Additionally, guidelines, comments, editorials, duplicate publications, studies that were not systematic reviews, and abstracts without full-text availability were also excluded from the analysis.

### Search strategy

Five electronic databases, including Cochrane Library, Embase, PubMed, Scopus, and Web of Science, limited to the English language, were searched systematically until October 15, 2024. To capture grey literature, WorldCat and the first 100 hits of Google Scholar were also explored. The search strategy comprised terms presented in Table 1.

### Data extraction and synthesis

Duplicate records were removed using Mendeley Reference Manager and two independent reviewers (SA and JK) performed the preliminary screening of titles and abstracts in accordance with the eligibility criteria. Full-text relevant records were retrieved and screened. During the screening stage, text mining was also conducted using the SWIFT-Review software (Sciome LLC, NC, USA). This program automatically groups abstracts related to comparable subjects using machine learning methods [14]. We employed AI algorithms to search, categorize, and prioritize large number of primary studies during the screening stage using SWIFT-Review software [15]. However, the final decisions on inclusion were based on human judgment. Discrepancies among the independent reviewers were resolved using the Delphi methodology during each phase [16].

The characteristics of included systematic reviews was extracted by two independent authors (SA and AK) using a standard Joanna Briggs Institute extraction form [17], which included the following details:

- Study characteristics: the first author's name, year of publication, number of databases searched, number and types of included studies, date of search, quality assessment tool, and results of the meta-analysis.

- AI techniques employed: artificial neural network, machine learning algorithms, deep learning method, or convolutional neural network.

- The types of caries: proximal, root, occlusal, or other parts of tooth.

**Table 1. Search strategy and number of records retrieved from each database on Oct 15, 2024.**

| Database | Search line | Number of retrieval records |
|---|---|---|
| PubMed | ("Artificial Intelligence"[MeSH Terms] OR "Support Vector Machine"[MeSH Terms] OR "Deep Learning"[MeSH Terms] OR "neural networks, computer"[MeSH Terms] OR ("Machine Learning"[MeSH Terms] OR "Unsupervised Machine Learning"[MeSH Terms] OR "Supervised Machine Learning"[MeSH Terms]) OR "diagnosis, computer assisted"[MeSH Terms] OR "Electronic Data Processing"[MeSH Terms] OR "Convolutional Neural Network"[Text Word] OR "CNN"[Text Word]) AND ("Dental Caries"[MeSH Terms] OR "Root Caries"[MeSH Terms]) | 253 |
| EmBase | ('artificial intelligence'/exp OR 'support vector machine'/exp OR 'deep learning'/exp OR 'artificial neural network'/exp OR 'ann (artificial neural network)' OR 'ann analysis' OR 'ann approach' OR 'ann method' OR 'ann methodology' OR 'ann methods' OR 'ann model' OR 'ann modeling' OR 'ann modelling' OR 'ann models' OR 'ann output' OR 'ann technique' OR 'ann techniques' OR 'ann training' OR 'anns (artificial neural networks)' OR 'algorithmic neural network' OR 'artificial nn' OR 'artificial nns' OR 'artificial neural network' OR 'artificial neural networks' OR 'computational neural network' OR 'computer neural network' OR 'computer neural networks' OR 'computerized neural network' OR 'connectionist model' OR 'connectionist network' OR 'connectionist neural network' OR 'connectionist system' OR 'mathematical neural network' OR 'neural network (artificial)' OR 'neural network (computer)' OR 'neural network algorithm' OR 'neural network model' OR 'neural networks (computer)' OR 'neural networks, computer' OR 'machine learning'/exp OR 'convolutional neural network'/exp OR 'cnn (convolutional neural network)' OR 'cnns (convolutional neural networks)' OR 'convnet' OR 'convoluted neural network' OR 'convolution neural network' OR 'convolutional anns' OR 'convolutional nn' OR 'convolutional artificial neural network' OR 'convolutional deep neural network' OR 'convolutional neural network' OR 'convolutionary neural network' OR 'deep convolutional neural network') AND ('dental caries'/exp OR 'dental caries' OR 'white spot lesion'/exp OR 'white spot lesion') | 352 |
| Scopus | #1 TITLE-ABS-KEY ("Artificial intelligence" OR ai OR "support vector machine" OR "Deep learning" OR "Machine based algorithm*" OR "Neural network*" OR nn OR ann OR cnn OR "Machine learning")<br>#2 TITLE-ABS-KEY (dental AND (caries OR cavit* OR decay* OR "White Spot*" OR lesion*))<br>#3 TITLE-ABS-KEY (caries AND (detection OR diagnosis))<br>#4 TITLE-ABS-KEY (t**th AND (lesion* OR caries))<br>#5 TITLE-ABS-KEY (carious AND (lesion* OR dentin*))<br>#6 TITLE-ABS-KEY (bitewing OR periapical OR pa OR panoramic OR opg OR "Panoramic Radiograph*" OR orthopantomography OR "Cone Beam Computed Tomography" OR cbct OR "Cone Beam CT" OR "Dental CT" OR "Intraoral Photographic Image*" OR "Photographic Image*" OR "Near Infrared Light Transillumination" OR nilt OR "Dental PRE/2 Tomography")<br>#7: #2 OR #3 OR #4 OR #5<br>#8: #1 AND #6 AND #7 | 426 |
| Web of Science | #1 TS=("Artificial intelligence" OR AI OR "Support Vector Machine" OR "Deep learning" OR "Machine based algorithm*" OR "Neural network*" OR NN OR ANN OR CNN OR "Machine learning")<br>#2 TS=(Dental AND (Caries OR Cavit* OR Decay* OR "White Spot*" OR Lesion*))<br>#3 TS=(Caries AND (Detection OR Diagnosis))<br>#4 TS=(T**th AND (Lesion* OR Caries))<br>#5 TS=(Carious AND (Lesion* OR Dentin*))<br>#6 TS=(bitewing OR periapical OR pa OR panoramic OR opg OR "Panoramic Radiograph*" OR orthopantomography OR "Cone Beam Computed Tomography" OR cbct OR "Cone Beam CT" OR "Dental CT" OR "Intraoral Photographic Image*" OR "Photographic Image*" OR "Near Infrared Light Transillumination" OR nilt OR "Dental PRE/2 Tomography")<br>#7: #2 OR #3 OR #4 OR #5<br>#8: #1 AND #6 AND #7 | 277 |
| Cochrane Reviews | ("artificial intelligence") AND ("dental caries")" (Word variations have been searched) | 1 |
| WorldCat | ti:("Artificial intelligence" OR "support vector machine" OR "Deep learning" OR "Neural network*" OR "Machine learning") AND ("Dental Caries" OR "Dental Cavit*" OR "Dental Decay*" OR "Dental White Spot*" OR "Carious Lesion*" OR "Caries Detection") | 119 |
| Google Scholar | ("Artificial intelligence" OR "support vector machine" OR "Deep learning" OR "Neural network*" OR "Machine learning") AND ("Dental Caries" OR "Dental Cavit*" OR "Dental Decay*" OR "Dental White Spot*" OR "Carious Lesion*" OR "Caries Detection") limited to review articles | First 100 hits |

- Dental imaging modalities used: bitewing, periapical, panoramic radiographs, cone beam computed tomography, and intraoral photographic images.

- Measures to be pooled: sensitivity, specificity, log diagnostic odds ratio, area under curve.

The retrieved data were analyzed and qualitatively summarized to assess the diagnostic accuracy of artificial intelligence for detecting dental caries across a range of imaging modalities. Overlapping studies within the included reviews were handled by creation of a citation matrix and calculation of corrected covered area (CCA) using the ccaR package of R software (R Foundation for Statistical Computing, Vienna, Austria) [18].

### Meta-analysis

Original studies provided data on True Positive (TP), False Positive (FP), False Negative (FN), and True Negative (TN) counts included in the meta-analysis. We also used a backward calculation of TP, TN, FP, and FN using: 1) Prevalence (TP+FN/ Total sample size), sensitivity, specificity, and total sample size, 2) Prevalence, positive likelihood ratio, negative likelihood ratio, and total sample size, or 3) Sensitivity, specificity, total number of positive and total number of negative test results [19].

A random effect model was used to combine sensitivity, specificity, log diagnostic odds ratio, AUC estimates of diagnostic tests among the original studies included in the systematic reviews. The Wald test was used to calculate the confidence intervals (CIs). The Galbraith plot was employed to assess heterogeneity and detect potential outliers. To quantify the potential effect of these outliers on the estimation of the pooled variables, leave-one-out meta-analyses were carried out. The regression-based Egger test and nonparametric rank correlation Begg test were employed for assessment of small-study effects. Nonparametric trim-and-fill analysis was employed to evaluate the number of studies potentially missing from the meta-analysis. To assess post-test probabilities, the Fagan nomogram was used [20]. The meta-analysis and visualizations were carried out using Stata 18 (StataCorp, College Station, TX, USA) and the mada, MetaDTA, and nsROC packages of R software. We employed an online diagnostic test calculator hosted by the University of Illinois at Chicago to draw Fagan nomogram. Interaction between imaging method and AI algorithm for sensitivity, specificity, and log diagnostic odds ratio assessed by random forests model (a machine learning algorithm) using a metaforest package.

### Quality and risk of bias assessment

The quality and risk of bias of the included systematic reviews were assessed by two independent researchers (SA and MY) using the AMSTAR 2 (A MeaSurement Tool to Assess systematic Reviews) tool [21]. The risk of bias in the original studies included in the systematic reviews eligible for meta-analysis was evaluated using the QUADAS-2 tool [22]. Disagreements regarding the quality evaluation were resolved by a Delphi technique [16].

## Results

### Characteristics of the included reviews

The initial search yielded a total of 1120 studies (Table 1). Following the removal of duplicates and the title-abstract screening process, 25 records met the criteria for full-text review. Ultimately, after excluding 11 records (Table 2), the remaining 14 records which met the eligible criteria were included (Fig 1). Twelve of the included records were systematic reviews and two were systematic reviews with meta-analyses (Table 3). The reviews covered a range of publication years, with 2 conducted prior to 2022, 6 conducted in 2022, 1 conducted in 2023, and 5 conducted in 2024. The original studies included in the reviews spanned from 1984 to 2023, providing a broad range of evidence.

The included reviews encompassed a range of AI techniques, prominently CNN, employed for dental caries detection. These reviews assessed the performance of AI algorithms in detecting dental caries using diverse dental imaging modalities, including intraoral photographic images, periapical radiographs, bitewing radiographs, CBCT images,

**Table 2. Excluded systematic reviews and the reason for exclusion.**

| Study | Year of publication | Title | Reason for exclusion |
|---|---|---|---|
| Chifor et al. [23] | 2022 | Automated diagnosis using artificial intelligence a step forward for preventive dentistry: A systematic review. | Conditions not concerning dental caries diagnosis |
| Hegde et al. [24] | 2022 | Deep learning algorithms show some potential as an adjunctive tool in caries diagnosis. | Editorial |
| Alqutaibi et al. [25] | 2023 | Artificial intelligence (AI) as an aid in restorative dentistry is promising, but still a work in progress. | Editorial |
| Fatima et al. [26] | 2023 | A systematic review on artificial intelligence applications in restorative dentistry. | Duplicate publication of Revilla-León M, et al. [27] |
| Futyma-Gabka et al. [28] | 2021 | The use of artificial intelligence in radiological diagnosis and detection of dental caries: a systematic review | Critically low quality (based on AMSTAR 2) |
| Musri et al. [29] | 2021 | Deep learning convolutional neural network algorithms for the early detection and diagnosis of dental caries on periapical radiographs: A systematic review | Critically low quality (based on AMSTAR 2) |
| Singh et al. [30] | 2022 | Progress in deep learning-based dental and maxillofacial image analysis: A systematic review | Critically low quality (based on AMSTAR 2) |
| Forouzeshfar et al. [31] | 2023 | Dental caries diagnosis using neural networks and deep learning: a systematic review | Critically low quality (based on AMSTAR 2) |
| Bhat et al. [32] | 2023 | A comprehensive survey of deep learning algorithms and applications in dental radiograph analysis | Critically low quality (based on AMSTAR 2) |
| Al-Namankany et al. [33] | 2023 | Influence of artificial intelligence-driven diagnostic tools on treatment decision making in early childhood caries: A systematic review of accuracy and clinical outcomes | Conditions not concerning dental caries diagnosis |
| Alam et al. [34] | 2024 | Applications of artificial intelligence in the utilisation of imaging modalities in dentistry: A systematic review and meta-analysis of in-vitro studies | Conditions not concerning dental caries diagnosis |
| Pecorari et al. [35] | 2024 | The use of artificial intelligence in the diagnosis of carious lesions: Systematic review and meta-analysis | Not peer-reviewed and a preprint article |

near-infrared-light transillumination, panoramic radiographs, and others. Quality assessment among the included systematic reviews exhibited predominantly moderate quality (Table 4).

The citation matrix of primary studies included in the systematic reviews is presented in S1 Table. The CCA_Proportion was 0.07 and CCA_Percentage was 6.90, showing moderate overlap. The pairwise CCA presented in S1 Fig shows which combinations of paired reviews had the highest overlap.

## Meta-analysis of eligible original studies

Among the fourteen systematic reviews included, 137 relevant original studies were identified (S1 Table). Only 20 original articles reported numbers of TP, TN, FP, and FN and were included in the meta-analysis (Table 5). Quality assessment of diagnostic accuracy studies among the included original studies revealed the lowest risk regarding index test applicability concerns and the highest risk regarding index test risk of bias (Fig 2).

Of the 29423 diagnostic tests analyzed from the results retrieved from the 20 articles (Fig 3), the pooled sensitivity was 0.85 (95% CI: 0.83 to 0.93), specificity was 0.90 (95% CI: 0.85 to 0.95), and log diagnostic odds ratio was 4.37 (95% CI: 3.16 to 6.27) (diagnostic odd ratio: 70.9 (95% CI: 44.9 to 111.9)) as shown in Fig 4. Results of heterogeneity assessments related to sensitivity, specificity, and log diagnostic odds ratio among the included studies are provided in Fig 5. The studies of Zadrozny L, 2022 [64], Park EY, 2022 (Faster R-CNN) [51], and De Araujo Faria V, 2021 [56], were outliers for sensitivity, specificity, and log diagnostic odds ratio, respectively. Results of the leave-one-out meta-analysis are shown in Fig 6. When omitting each study,

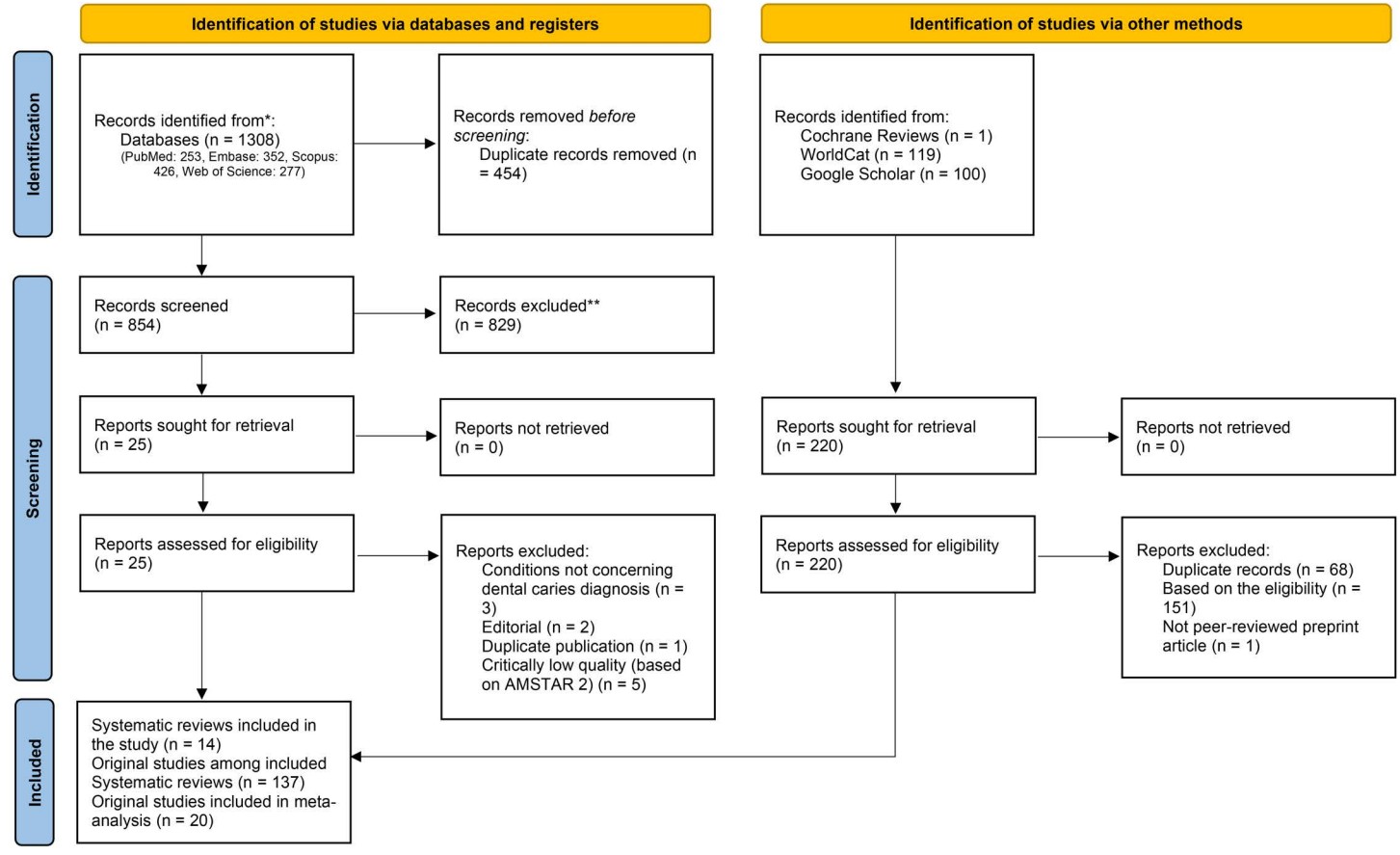

**Fig 1. Systematic review flowchart according to PRISMA 2020.**

results changed minimally, e.g., 0.01 in sensitivity and specificity and the first decimal place in log diagnostic odds ratio. The results of the regression-based Egger test and nonparametric rank correlation Begg test for assessment of small-study effects are provided in Table 6. The nonparametric trim-and-fill analysis of publication bias are in Table 6 and Fig 7. Nonparametric trim-and-fill analysis estimated 6 unpublished studies estimated for sensitivity and log diagnostic odds ratio. The meta-analysis for non-parametric ROC curves are presented in Fig 8. The area under the pooled ROC curve is 0.867. Positive and negative likelihood ratios were 10.443 (95% CI: 7.505 to 14.531) and 0.168 (95% CI: 0.138 to 0.205), respectively. Total number of tests were 29424, TP were 6836, and FN were 1209. Hence the prevalence of dental caries was 27.3%. The positive post-test probability was 79% and negative post-test probability was 6% (Fig 9). The interaction between imaging method and AI algorithm for sensitivity, specificity, and log diagnostic odds ratio showed in Fig 10. Finally, Fig 11 provides a summary of meta-analysis performance, along with visual evaluations of threshold effect, hierarchical summary ROC curve and heterogeneity of data.

## Discussion

The core question in this umbrella review and meta-analysis was "Are AI algorithms effective for the detection of dental caries?" Given the global burden of caries and the exponential growth of AI in diagnostics, the study is timely and necessary in order to inform clinicians, scientists and other stakeholders on the effectiveness of this emerging technology.

In this umbrella review, 14 systematic reviews were included (Table 3) and 12 systematic reviews excluded (Table 2). Among the included systematic reviews only 2 conducted a meta-analysis (Table 4, item 11), most likely due to

**Table 3. Data extracted from the included systematic reviews.**

| Study | Aim | No. of data-bases, included studies | Types of included studies | Date range of included studies | Quality assess-ment tool | Meta-analysis | AI techniques | The types of caries | Dental imaging modalities used | Findings |
|---|---|---|---|---|---|---|---|---|---|---|
| Talpur S et al. [36], 2022 | Diagnosis of DC with ML | 4, 12 | Clinical studies and con-ference papers | 2008–22 | Tool devel-oped by the authors | No | DL with algorithms CNN and ANN, Hidden layer propagation, SVM, Random Forest, Logistic Regression, Feedforward propagation and Feedback propaga-tion CNN- and ANN based research | Proximal, root, and occlusal | Different types of imaging modalities | Acc: 69.6–99.0% |
| Reyes LT et al. [37], 2022 | Diagno-sis and prognostic prediction of DC with ML | 5, 15 (10 articles related to the aim of present review) | Cross-sectional and lon-gitudinal studies | 2008–20 | QUADAS-2 | No | ANNs, J48, Random Tree, Random Forest, SVM, and Naïve Bayes | Proximal, occlusal, and smooth free surface | Conventional or digital radio-graphic, oral photographs, and NILT | AUC: 74.5–98.7% |
| Moharrami M et al. [38], 2023 | Diagnosis of DC with AI | 3, 19 | Clinical studies and con-ference papers | 2019–22 | Modified version of QUADAS-2 | No | CNNs, Mask R-CNN, SVM, Decision Tree, SSD-MobileNetV2, Faster R-CNN, Customized CNNs, Cus-tom UCDA, YOLO-based CNN, ResNet architectures, Shufflenet V2, DenseNet, ResNext, RetinaNet, Squeeze Net, RDFNet, VGG NET-16, VGG NET-19, InceptionV3, SSD-ConvNet, DentalNet | Occlusal and smooth free surface | Oral photographs | F1-scores: classification task: 68. 3–94.3%, and detection task: 42.8–95.4% |
| Prados-Privado M et al. [39], 2020 | Diagnosis of DC with DL | 3,13 | In vitro and clinical studies and con-ference papers | 2008–20 | Cochrane RoB | No | ResNet architectures, Res-Next50, ANN, CNN, DNN Mask R-CNN, Feedforward NN, RT and DCT, FCNN, MLP, Back propagation NN | Proximal, occlusal, pre-cavitated, and initial caries | PA, NILT, bite-wing, panoramic, radiovisiography, oral photographs | Acc: 68. 6–99.0%, Precision: 61.5–98.7%, AUC: 74.0–97.1% |
| Mohammad-Rahimi H et al. [40], 2022 | Diagnosis of DC with DL | 4, 42 | Clinical and in vitro studies | 2015–21 | Modified version of QUADAS-2 | No | Customized CNN structures, Transfer learning models, MLP, Auto-encoders (e.g., U-Net), One-stage object detectors (e.g., YoLo) or Two-stage object detectors (e.g., Faster R-CNN) | NA | Oral photo-graphs, PA, bite-wing, NILT, OCT, panoramic, and CBCT | Acc: 68.0–99.0% |
| Revilla-León M et al. [27], 2022 | Diagnostic perfor-mance of AI in restorative dentistry | 5, 34 (29 articles related to the aim of present review) | Clinical and in vitro studies | 1984-2020 | JBI critical appraisal checklist | No | Expert systems, Regres-sion analysis, Fuzzy logic learning, Perceptron NN, MLP, Back propagation NN, CNNs, ANNs, Decision tree learning | NA | PA, bitewing, oral photographs, NILT, and fiber optic displace-ment sensor | Acc: 76.0–88.3%, Se: 73.0–90.0%, Sp: 61.5–93.0% |

*(Continued)*

| Study | Aim | No. of databases, included studies | Types of included studies | Date range of included studies | Quality assessment tool | Meta-analysis | AI techniques | The types of caries | Dental imaging modalities used | Findings |
|---|---|---|---|---|---|---|---|---|---|---|
| Khanagar SB et al. [41], 2022 | Diagnosis and prediction of DC with AI | 7, 34 (24 articles related to the aim of present review) | No limitation for the type of study | 2018–22 | QUADAS-2 | No | DCNNs, CNNs, ANNs | NA | PA, NILT, bitewing, oral photographs, panoramic, OCT, and micro-CT images | Acc: 73.3–98.8%, Se: 59.0–98.9%, Sp: 68.9–98.2%, AUC: 74.0–98.7% |
| Khanagar SB et al. [42], 2022 | Application of AI in pediatric dentistry | 7, 20 (1 articles related to the aim of present review) | No limitation for the type of study | 2021 | QUADAS-2 | No | CNNs, ANNs | NA | Oral photographs | Se: 88.1–96.0%, Sp: 97.0–99.0% |
| Khanagar SB et al. [43], 2021 | Application of AI in dentistry | 7, 43 (4 articles related to the aim of present review) | No limitation for the type of study | 2008–20 | QUADAS-2 | No | DCNNs, CNNs, ANNs | NA | PA, bitewing, and NILT | AUC: 74.0–89.0% |
| Zanini LGK et al. [44], 2024 | Application of AI in caries detection, classification, and segmentation in X-ray images | 4, 42 | No limitation for the type of study | 2016–23 | Tool developed by the authors | No | IP, CSL, Inception V3, Hybrid graph-cut segmentation, CNN-SVM, AlexNet, VGG19, VGG16, DenseNet-121, U-Net, ResNet, GoogLeNet, MobileNet V2, Hybrid NN, XNet, SegNet, 3-layer CNN, Network MI-DCNNE, Mask R-CNN, Faster R-CNN, YOLOv3, CariesNet, ResNeXt, Deep gradient–based LeNet, EfficientNet-B0 | NA | PA, bitewing, x-ray, panoramic, and CBCT | NA |
| Ndiaye AD et al. [45], 2024 | Application of AI in caries detection, classification, and segmentation from radiographic databases | 4, 35 | Radiographic human studies | 2018–22 | QUADAS-AI | No | CNNs, SVM, Random Forest, Naive Bayes, K-nearest neighbors, auto-encoder, semi-supervised learning, support tensor machine, network-in-network, Transfer learning | NA | PA, bitewing, CBCT, and panoramic | Acc: 70.8–99.1%, Se: 41.6–99.4%, Sp: 66.3–98.9% |

*(Continued)*

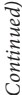

| Study | Aim | No. of data-bases, included studies | Types of included studies | Date range of included studies | Quality assess-ment tool | Meta-analysis | AI techniques | The types of caries | Dental imaging modalities used | Findings |
|---|---|---|---|---|---|---|---|---|---|---|
| Ammar N et al. [46], 2024 | Diagnosis of DC with AI on bite-wing radio-graphs | 5, 14 (5 articles included in meta-analysis) | No limitation for the type of study | 2020–23 | QUADAS-2 | Yes | AlexNet, ResNet, Incep-tion, U-Net, Faster R-CNN, YOLOv3 | Proximal | Bitewing | Odds Ratio: 55.8 (95% CI=28.8–108.3), Se: 76.0–94.0%, Sp: 75.0–96.0% |
| Rokhshad R et al. [47], 2024 | Application of AI in pediatric dentistry | 8, 33 (2 articles related to the aim of present review) | Clinical studies | 2008–23 | QUADAS-2 | Yes | ANNs | NA | Oral photographs | Acc: 78.0%, Se: 88.1–96.0%, Sp: 97.0–99.0% |
| Albano D et al. [48], 2024 | Detection of DC with AI | 5, 20 (19 articles related to the aim of present review) | No lim-itation for the type of study | 2008–22 | QUADAS-2 | No | DCNNs, CNNs, ANNs | NA | PA, bitewing, orthopantomog-raphy | Acc: 73.0–98.0%, Se: 44.0–86.0%, Sp: 85.0–98.0%, preci-sion: 50.0–94.0%, AUC: 84.0–98.0%, F1-score: 64.0–92.0% |

Abbreviations: DC, Dental Caries; ML, Machine Learning; DL, Deep Learning; ECC, Early Childhood Caries; RoB, Risk of Bias; CNN/ConvNet, Convolutional Neural Net-work; ANN, Artificial Neural Network; SVM, Support Vector Machine; FCNN, Fully Convolutional Neural Network; FCM, Fuzzy C-Means algorithm; RT, Radon transformation; DCT, Discrete Cosine Transformation; ResNet, Residual Network; NN, Neural Network; R-CNN, Region-based Convolutional Neural Network; SSD, Single Shot Detector; UCDA, Unified Caries Detection and Assessment; DCNN, Deep Convolutional Neural Network; RDFNet, RGB-D Fusion Network; VGG, Visual Geometry Group; LSTM, Long Short-Term Memory; MLP, Multi-Layer Perceptron; MPCA, Modified Principal Component Analysis; IP, Image Processing; CSL, Classical Supervised Learning; MI-DCNNE, Multi-Input Deep Convolutional Neural Network Ensemble; NA, Not Available; PA, Peri-Apical; CBCT, Cone-beam computed tomography; CT, Computed Tomography scans; OCT, Optical Coherence Tomography; NILT, Near-Infrared-Light Transillumination; Acc, Accuracy; Se, Sensitivity; Sp, Specificity; AUC, Area Under Curve.

Table 4.  Quality assessment of each included systematic review according to AMSTAR 2.

| Systematic Review | 1 | 2 | 3 | 4 | 5 | 6 | 7 | 8 | 9 | 10 | 11 | 12 | 13 | 14 | 15 | 16 | Overall Quality |
|---|---|---|---|---|---|---|---|---|---|---|---|---|---|---|---|---|---|
| Reyes LT et al. [37] | Yes | Yes | Yes | Yes | Yes | Yes | Yes | Yes | Yes | No | No MA | No MA | Yes | Yes | No MA | Yes | High |
| Mohammad-Rahimi H et al. [40] | Yes | Yes | Yes | Yes | Yes | Yes | Yes | Yes | Yes | No | No MA | No MA | Yes | Yes | No MA | Yes | High |
| Khanagar SB et al. [41] | Yes | PY | Yes | Yes | Yes | Yes | Yes | Yes | Yes | No | No MA | No MA | Yes | Yes | No MA | Yes | High |
| Ndiaye AD et al. [45] | Yes | Yes | Yes | Yes | Yes | Yes | Yes | Yes | Yes | No | No MA | No MA | Yes | Yes | No MA | Yes | High |
| Moharrami M et al. [38] | Yes | Yes | Yes | Yes | Yes | No | Yes | Yes | Yes | No | No MA | No MA | Yes | Yes | No MA | Yes | Moderate |
| Prados-Privado M et al. [39] | Yes | PY | Yes | Yes | Yes | Yes | Yes | Yes | Yes | No | No MA | No MA | Yes | Yes | No MA | Yes | Moderate |
| Revilla-León M et al. [27] | Yes | PY | Yes | Yes | Yes | Yes | Yes | Yes | Yes | No | No MA | No MA | Yes | No | No MA | Yes | Moderate |
| Khanagar SB et al. [42] | Yes | No | Yes | PY | Yes | Yes | Yes | Yes | Yes | No | No MA | No MA | Yes | Yes | No MA | Yes | Moderate |
| Khanagar SB et al. [43] | Yes | PY | Yes | Yes | Yes | Yes | Yes | Yes | Yes | No | No MA | No MA | Yes | Yes | No MA | Yes | Moderate |
| Zanini LGK et al. [44] | Yes | Yes | No | PA | Yes | Yes | Yes | Yes | Yes | No | No MA | No MA | Yes | Yes | No MA | Yes | Moderate |
| Ammar N et al. [46] | Yes | PY | Yes | Yes | Yes | Yes | Yes | Yes | PY | No | Yes | No | Yes | Yes | Yes | Yes | Moderate |
| Rokhshad R et al. [47] | Yes | PY | Yes | Yes | Yes | Yes | Yes | Yes | Yes | No | Yes | No | Yes | Yes | Yes | Yes | Moderate |
| Talpur S et al. [36] | Yes | PY | Yes | PY | No | No | Yes | Yes | PY | No | No MA | No MA | Yes | Yes | No MA | Yes | Low |
| Albano D et al. [48] | Yes | No | Yes | Yes | Yes | Yes | Yes | Yes | Yes | No | No MA | No MA | Yes | No | No MA | Yes | Low |

Abbreviations: PY, Partial Yes; MA, Meta-Analysis; Low, Low quality. Numbers at first row showed the items in AMSTAR 2. [21]

the failure of the original studies they included to report adequate details. We analyzed 29423 diagnostic tests, which resulted in a pooled sensitivity of 0.85 (95% CI: 0.83 to 0.93) and specificity of 0.90 (95% CI: 0.85 to 0.95) (Fig 4). Readers must note, the type of dental imaging strongly influences diagnostic performance. AI evaluating panoramic radiographs (with tooth overlap) operates under very different conditions than AI analyzing bitewing images (with clear approximal visibility). Pooling such results may reduce the clinical interpretability of the findings. To address this issue we conducted a subgroup meta-analysis for different imaging modalities with the results presented in Fig 4.

It is well-known that, the most reliable conclusions would come from reviews comparing studies conducted under the same diagnostic protocols and caries definitions. As showed in Table 5, caries detection method, reference standard and AI algorithm were the same among the studies included in the meta-analysis.

Nevertheless, Ammar N et al., [46] (2024) in a recent systematic review and meta-analysis assessed diagnostic performance of AI algorithms for caries detection on bitewing radiographs. Among 5 included studies, the pooled sensitivity and specificity were 0.87 (95% CI: 0.76 to 0.94) and 0.89 (95% CI: 0.75 to 0.96). Macey R, et al., (2021) [69] in a recent systematic review and meta-analysis, which included 67 studies reporting a total of 19590 tooth sites/surfaces, assessed the diagnostic accuracy of several visual classification systems for the detection and diagnosis of non-cavitated coronal dental caries. For all visual classification systems, the pooled sensitivity and specificity were 0.86 (95% CI 0.80 to 0.90) and 0.77 (95% CI 0.72 to 0.82) respectively. In another systematic review and meta-analysis, Iranzo-Cortés JE, et al., (2019) [70]examined the accuracy of caries diagnostic tools based on laser fluorescence in pre-cavitated carious lesions. For 655 nm light wavelength lasers, which included 25 studies, the pooled sensitivity and specificity were 0.77 (95% CI: 0.70 to 0.83) and 0.75 (95% CI: 0.69 to 0.80), respectively. For 405 nm light wavelength lasers, which included 13 studies, the pooled sensitivity and specificity were 0.81 (95% CI: 0.68 to 0.89) and 0.75 (95% CI: 0.62 to 0.85), respectively. The meta-analysis of Walsh T, et al., (2022) [5] which included 64 studies, reported pooled sensitivity and specificity for fluorescence-based devices 0.76 and 0.83, for analog and digital radiographs 0.50 and 0.89, for electrical conductance or impedance 0.83 and 0.72, and for transillumination and optical coherence tomography 0.76 and 0.82, respectively. Macey R, et al. (2020) [71] in a systematic review and meta-analysis of diagnostic test accuracy of fluorescence-based devices for diagnosis of enamel caries reported estimated sensitivity of 0.70 and specificity of 0.78 among 79 included studies.

**Table 5. Data extracted from the included original studies for meta-analysis.**

| Study | Dental imaging modality | Caries detection | Reference standard | AI algorithms (in detail) |
|---|---|---|---|---|
| Liu L [49] 2015 | Intraoral photographic images | Overall | Dental experts | CNNs (Mask R-CNN) |
| Kühnisch J [50] 2022 | Intraoral photographic images | Overall | A dental expert | CNNs (MobileNetV2) |
| | | Detection of cavitations | | CNNs (MobileNetV2) |
| Park EY [51] 2022 | Intraoral photographic images | Overall | A dental expert | CNNs (ResNet-18) |
| | | | | CNNs (ResNet-18 with U-net) |
| | | | | CNNs (Faster R-CNN) |
| | | | | CNNs (Faster R-CNN with U-net) |
| Vinayahalingam S[52] 2021 | Panoramic radiographs | Overall | Dental experts | CNNs (MobileNet V2) |
| Ezhov M [53] 2021 | CBCT | N/A | Dental and radiology experts | CNNs (U-net) |
| Devlin H [54] 2021 | Bitewing radiographs | Enamel | Dental experts | CNNs (AssistDent AI software) |
| Zheng L [55] 2021 | Periapical radiographs | Dentin | Dental experts | CNNs (VGG19) |
| | | | | CNNs (Inception V3) |
| | | | | CNNs (ResNet-18) |
| De Araujo Faria V[56], 2021 | Panoramic radiographs | Overall | Dental experts | ANNs |
| Oztekin F [57] 2023 | Panoramic radiographs | Overall | A dental expert | CNNs (EfficientNet-B0) |
| | | | | CNNs (DenseNet-121) |
| | | | | CNNs (ResNet-50) |
| Imak A [58] 2022 | Periapical radiographs | Overall | A dental expert | CNNs (MI-DCNNE) |
| Chen X [59], 2022 | Bitewing radiographs | Overall | 2 endodontic experts and 1 radiologist | CNNs (Faster R-CNN) |
| Li S [60] 2022 | Periapical radiographs | Overall | Dental experts | CNNs (VGG16) |
| | | | | CNNs (modified ResNet-18 backbone) |
| Fariza A [61] 2022 | Panoramic radiographs | Overall | Dental experts | CNNs (ResNet-18) |
| | | | | CNNs (ResNeXt50 32×4d) |
| Jayasinghe H [62] 2022 | Periapical radiographs | Overall | Dental experts | CNNs (ResNet-101) |
| Ari T [63] 2022 | Periapical radiographs | Overall | An oral and maxillofacial radiologist | CNNs (U-net) |
| Zadroˇzny L [64] 2022 | Panoramic radiographs | Overall | Dental experts | CNNs (Diagnocat) |
| Suttapak W [65] 2022 | Bitewing radiographs | Overall | An oral and maxillofacial radiologist | CNNs (ResNet-101) |
| Liu F [66] 2022 | Periapical radiographs | Overall | Dental experts | CNNs (VGG16) |
| | | | | CNNs (Inception V3) |
| | | | | CNNs (ResNet-50) |
| | | | | CNNs (DenseNet-121) |
| Estai M [67] 2022 | Bitewing radiographs | Overall | Dental experts | CNNs (Inception-ResNet-V2) |
| Bayraktar Y [68] 2022 | Bitewing radiographs | Overall | Dental experts | CNNs (YOLOv3) |

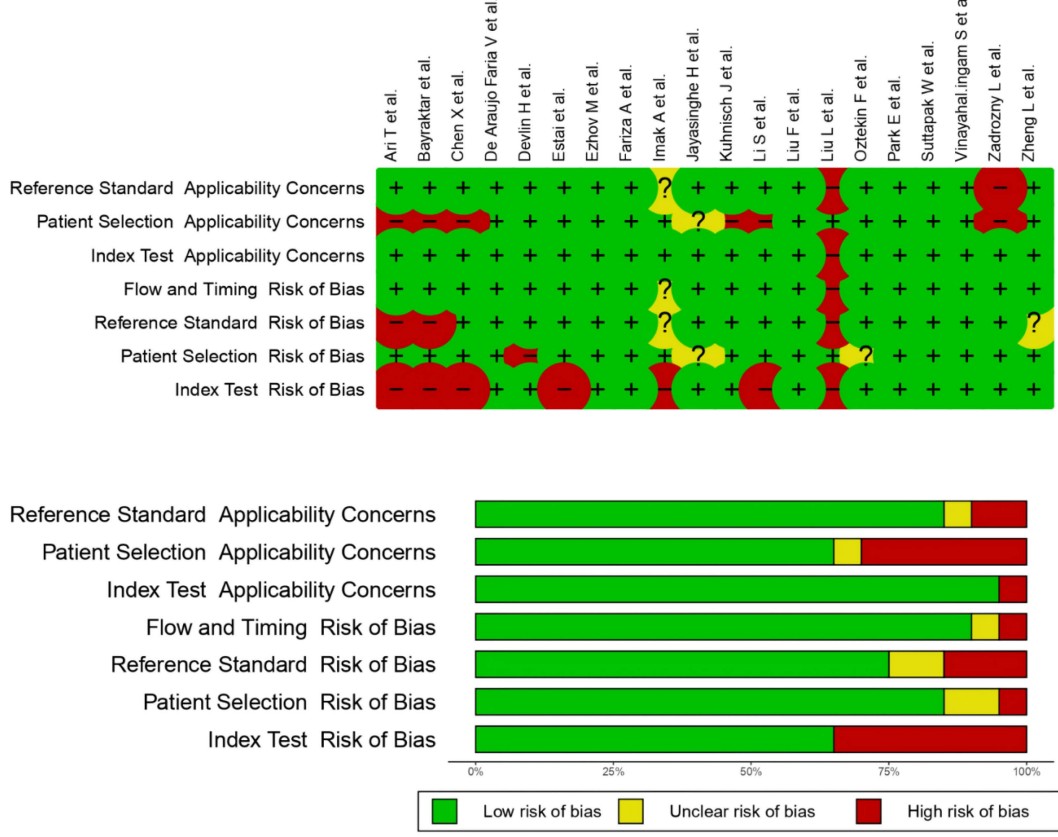

**Fig 2. Quality assessment of diagnostic accuracy studies among the included original studies for meta-analysis according to QUADAS-2.**

Among 29423 diagnostic tests analyzed in the present study, the prevalence estimate of dental caries was 27.3% (Fig 3). Results of the meta-analysis revealed a positive likelihood ratio of 10.443 (95% CI: 7.505 to 14.531), meaning that a positive test result is 8.47 time more likely to occur in someone who has caries, compared to someone who does not have caries, and a negative likelihood ratio of 0.168 (95% CI: 0.138 to 0.205), meaning that a negative test result is 0.17 time more likely to occur in someone who has caries compared to someone who does not have caries. The positive post-test probability was 79%, meaning that if a patient test is positive, there is a 79% chance they actually have caries, and negative post-test probability of 6%, meaning that if a patient test is negative, there is a 6% chance they actually have caries (Fig 9).

Sensitivity and specificity are a well-known pair of indicators for assessment of diagnostic test accuracy. There have been many efforts to combine the results of a diagnostic study into one single measure, for example the diagnostic odd ratio [72]. The results of a diagnostic odd ratio assessment ranges from 0 to infinity (with 1 as null value), where higher values indicate better diagnostic test performance [73]. In the present study the pooled log diagnostic odd ratio was 4.37 (diagnostic odd ratio: 70.9). This value provides a measure of how much likely a positive test result occurs in a person with dental caries compared to person without dental caries. Ammar N et al., (2024) [46] reported a pooled diagnostic odds ratio of 55.8 for AI-based caries detection on bitewing radiographs. In the study of Macey R, et al., (2021) [69] that included all visual caries classification systems, the pooled diagnostic odds ratio was 20.38. In another meta-analysis,

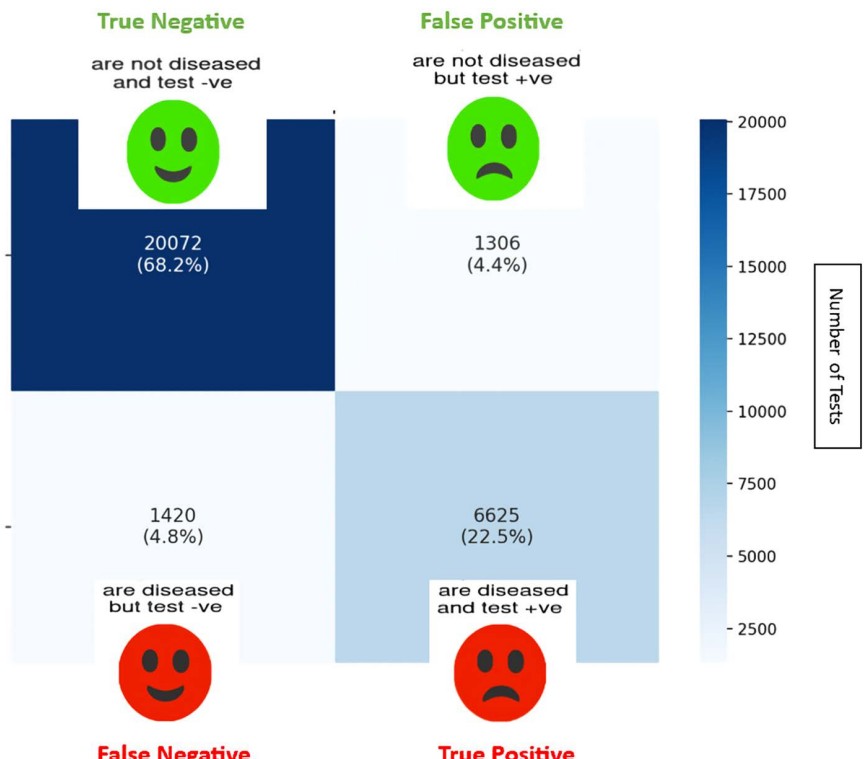

**Fig 3. Summary of the 29423 diagnostic tests included in the meta-analysis.**

Macey R, et al., (2020) [71] reported that the pooled diagnostic odds ratio for fluorescence-based enamel caries diagnostic devices was 14.1.

As a general rule, the AUC serves as an overall summary of diagnostic test performance. In the present study, the area under the pooled ROC curve for AI algorithms for the detection of dental caries was 0.86. This means that there is high probability that a randomly chosen individual with caries will have higher test score than a randomly chosen individual without disease. Iranzo-Cortés JE, et al., (2019) [70] reported that the area under the pooled ROC curve was 0.81 for 655 nm light wavelength lasers and was 0.80 for 405 nm light wavelength lasers.

The Egger's and Bag's test results were significant (p<0.05) for sensitivity, specificity, and log diagnostic odds ratio confirming that publication bias existed within the 20 original studies included in the meta-analysis. The nonparametric trim-and-fill analysis estimated 6 unpublished studies and presence of publication bias for sensitivity, and log diagnostic odds ratio. This publication bias may be related to difficulties related to the publication of innovative interdisciplinary high-tech research outcomes.

High levels of heterogeneity ($I^2 > 95$) were found regarding sensitivity, specificity, and log diagnostic odds ratio among the included studies. These levels of heterogeneity can be explained by diversity in imaging modalities (e.g., bitewing vs. panoramic radiographs), differences in AI algorithm architectures, and training dataset characteristics.

The main limitation of this umbrella review and meta-analysis include the fact that only 20 original studies were included in the meta-analysis quantitative data synthesis. The number of primary original research studies was 137, with only 20 (14.5%) original research articles reporting the necessary details of AI–based caries diagnostic test results including numbers TP, TN, FP, and FN involved in the meta-analysis. To facilitate future meta-analysis, authors are encouraged to report details of AI–based caries diagnostic tests including numbers of TP, TN, FP, and FN, all of which are essential for a meta-analysis.

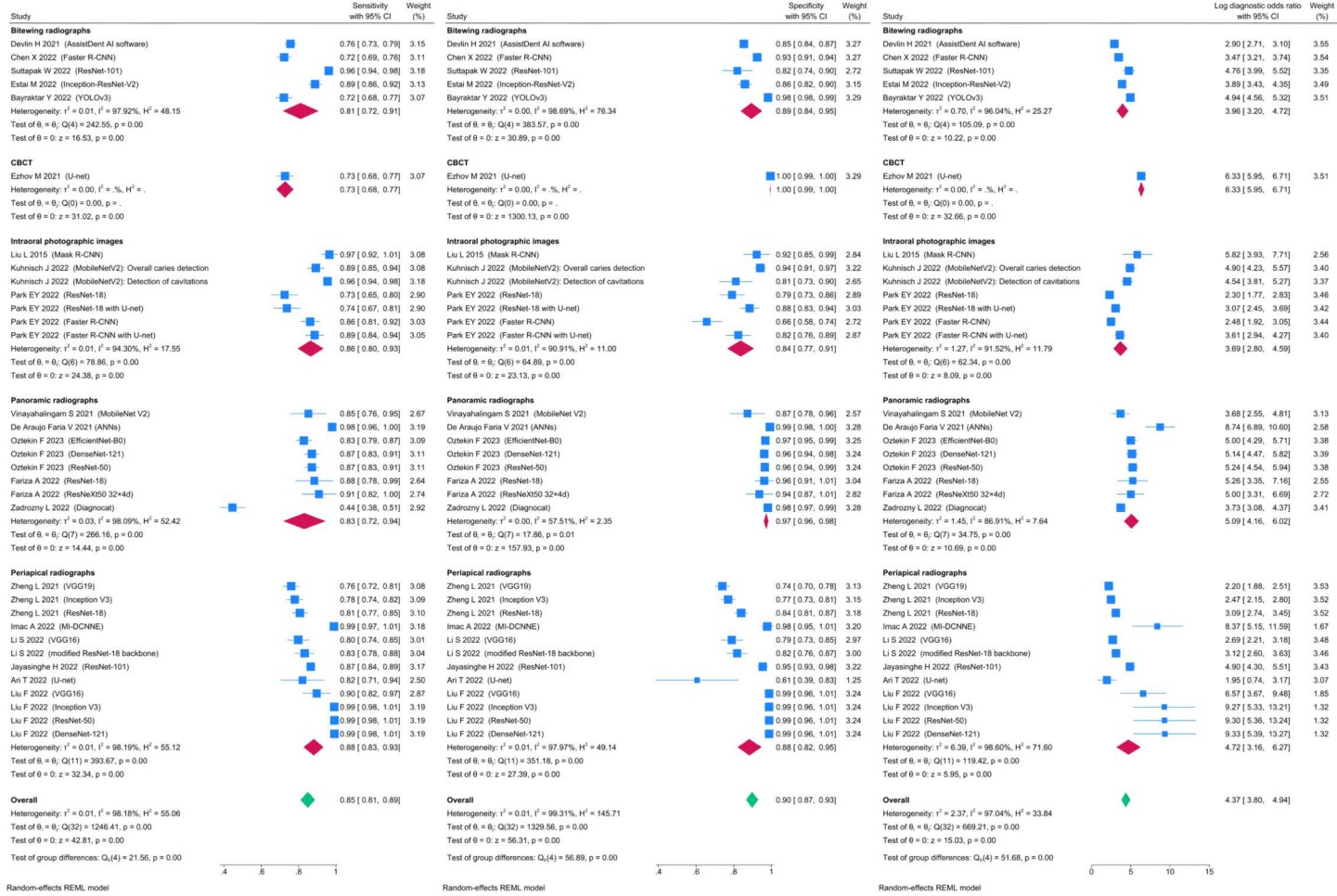

**Fig 4. Forest plots showing pooled sensitivity, specificity, and log diagnostic odds ratio among the included studies.** Test of sub-group differences for sensitivity, specificity, and log diagnostic odds ratio were significant (P<0.001).

The prevalence of dental caries among the 20 original research articles included in the meta-analysis was 27.3%, which is noticeably lower than the prevalence of dental caries in the real community suggesting a degree of selection bias among images datasets used for training and validation of AI algorithms. Future studies must involve the training and validation of AI algorithms using image datasets from a broader range demographic locations and healthcare settings to better reflect the real-world prevalence of dental caries.

We tried to conduct sub-group analysis according to the depth of the carious lesions and their location, for instance, enamel caries, dentine caries or root caries. Yet, among the majority of the original studies included, the AI algorithms were trained to detect dental caries and could not distinguish different depths and locations. Future research should focus on training AI algorithms to diagnose caries at different sites and at different depth of progression.

However, in future analyses, the most valuable evidence will likely come from studies that simulate real-world diagnostic conditions, including time pressure, clinician–patient interaction, and the integration of AI as part of the clinical workflow rather than in isolated image review [74].

In summary, this meta-analysis supports the use of AI in clinical practice for the detection of dental caries. However, AI algorithms are being developed and implemented rapidly and their accuracy for detection of dental

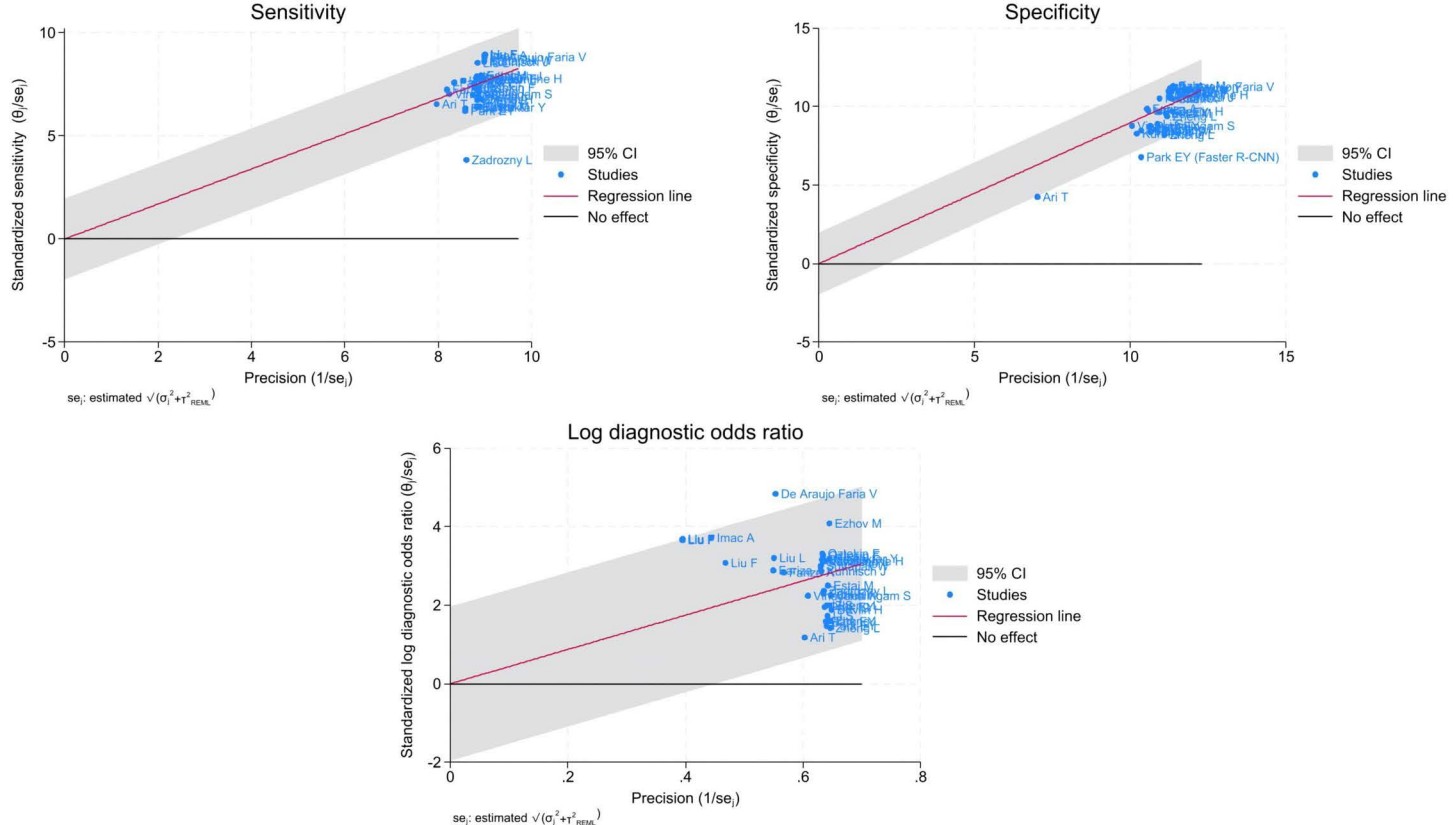

**Fig 5. Galbraith plots showing the results of heterogeneity assessments related to sensitivity, specificity, and log diagnostic odds ratio among the included studies.**

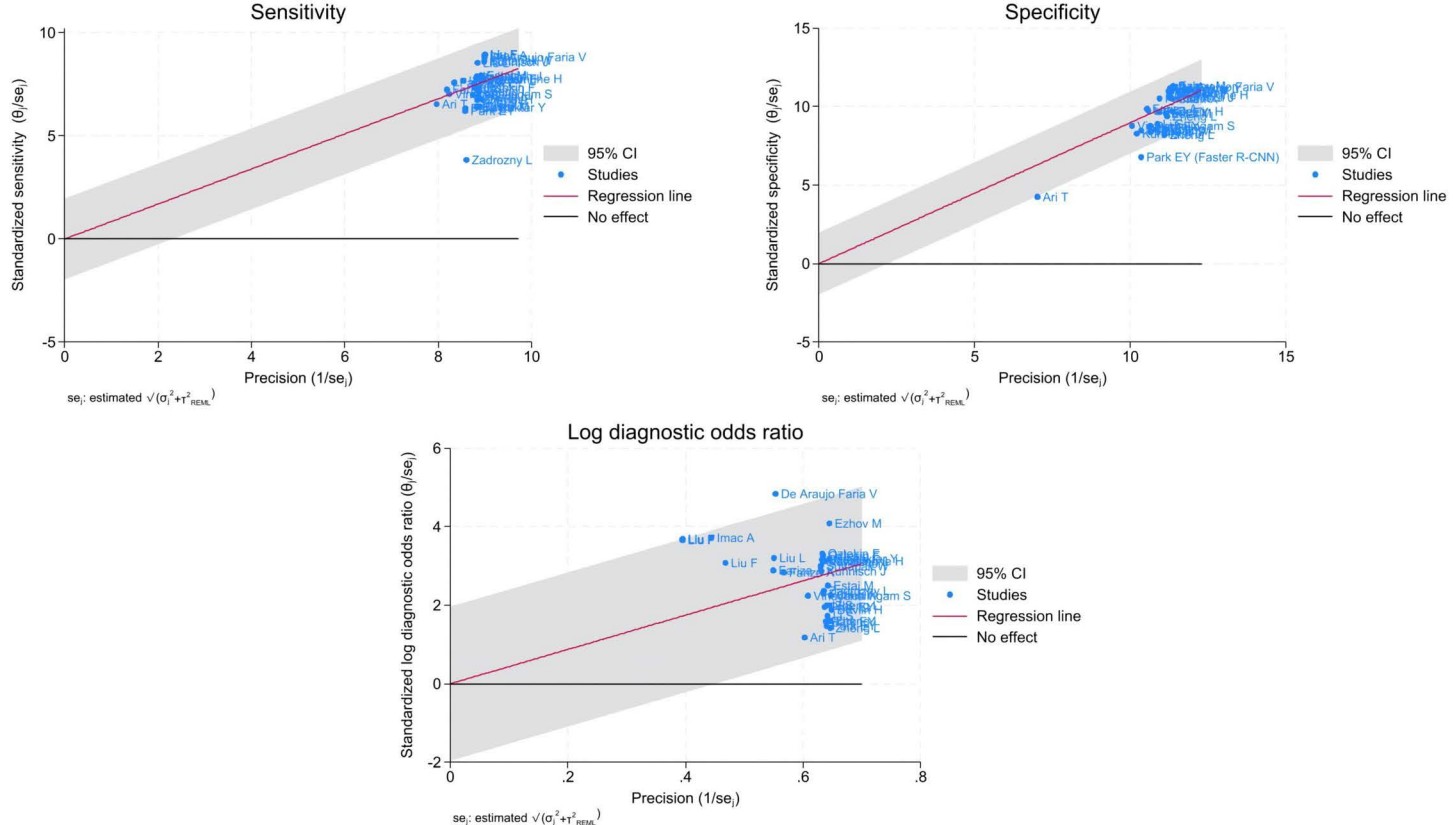

**Fig 6. Forest plots showing the results of the leave-one-out meta-analysis for sensitivity, specificity, and log diagnostic odds ratio among the included studies.**

**Table 6. Results of small-study effects related to publication bias and results of nonparametric trim-and-fill analysis, showing the impact of publication bias on the pooled estimates.**

| | Egger's Regression | Egger's test | Kend-alls Tau | Begg's test | Number of imputed studies | Observed pooled effect size | Observed+Imputed pooled effect size |
|---|---|---|---|---|---|---|---|
| **Sensitivity** | −3/201 | P=0.00137 | −0/338 | P=0.0058 | 6 | 0.850 (95% CI: 0.811 to 0.889) | 0.818 (95% CI: 0.778 to 0.858) |
| **Specificity** | −3/062 | P=0.0022 | −0/485 | P<0.001 | 0 | 0.898 (95% CI: 0.867 to 0.929) | 0.898 (95% CI: 0.867 to 0.929) |
| **Log diagnostic odds ratio** | 6/408 | P<0.001 | 0/481 | P<0.001 | 6 | 4.368 (95% CI: 3.798 to 4.938) | 3.913 (95% CI: 3.219 to 4.606) |

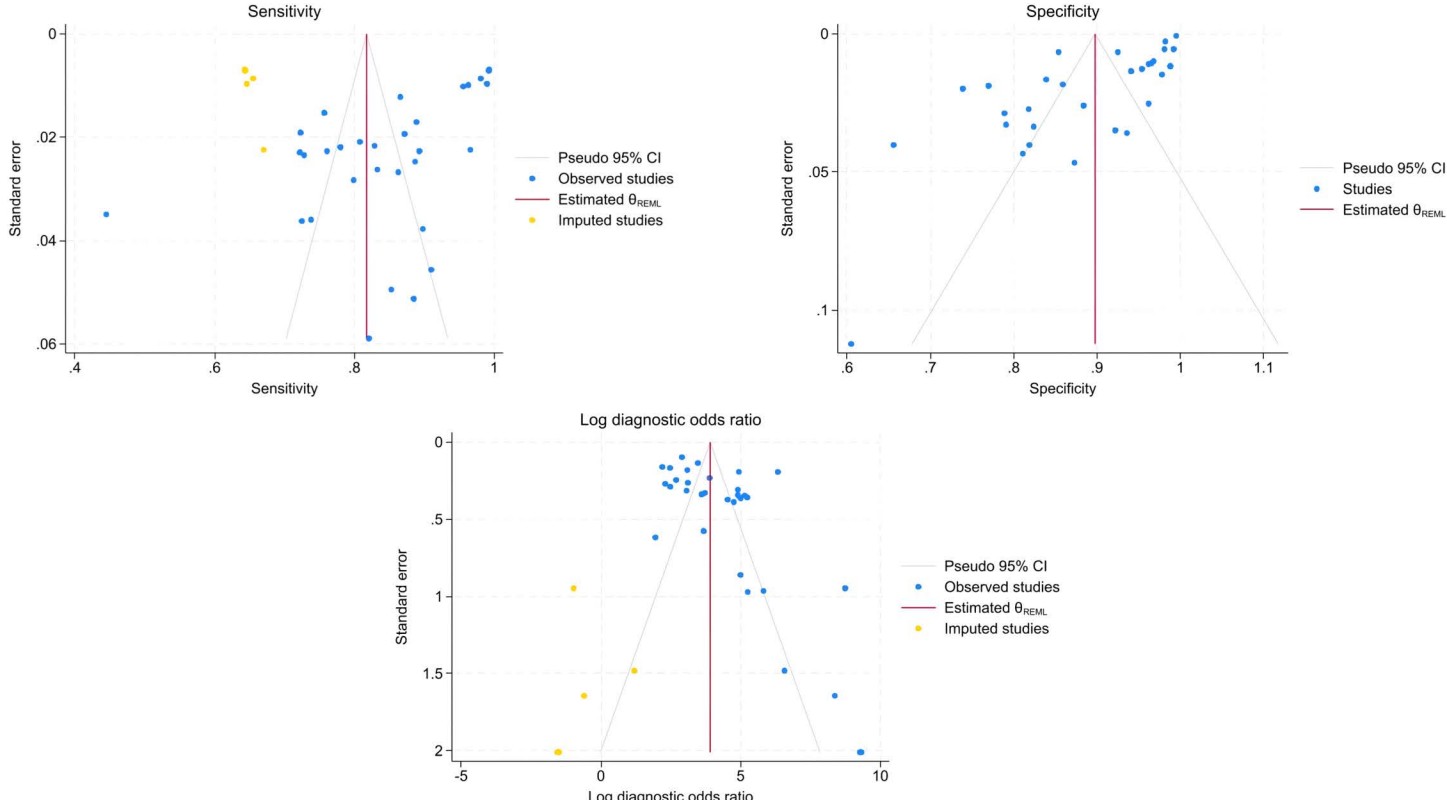

**Fig 7. Funnel plot showing the results of the nonparametric trim-and-fill analysis of publication bias related to sensitivity, specificity, and log diagnostic odds ratio among the included studies.**

caries is likely to increase in future. The majority of the included studies in the meta-analysis used CNN for image processing and detection of dental caries. This type of deep learning algorithm is a powerful type of feed-forward artificial neural network [75] that learns features automatically via filter optimization. The accuracy of the CNN model for detection of dental caries can be improved by dealing with large datasets and high-resolution inputs from diverse demographics (tooth type, age, sex, and ethnicity) and imaging equipment. Dental professionals could participate in public volunteer computing programs and potentially upload their radiographic images and donate their unused CPU and GPU cycles to develop powerful CNN models for detection of dental caries with high levels of accuracy [76].

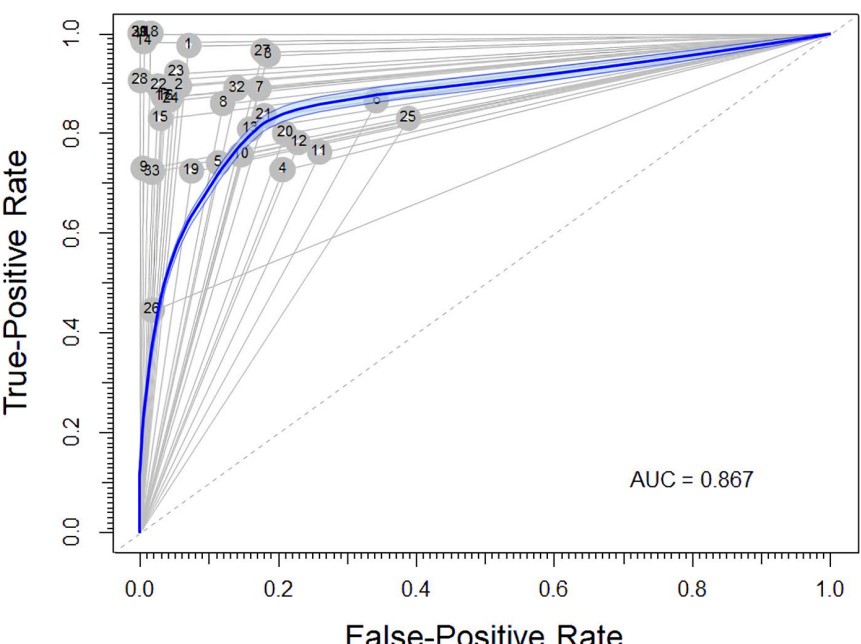

**Fig 8. Meta-analysis for non-parametric receiver operating characteristic (ROC) curve.** Studies numbered according to the order seen in forest plots in Fig 6.

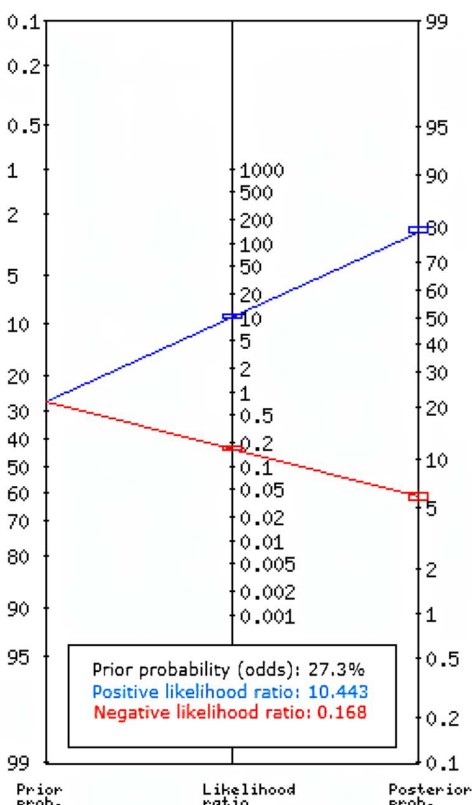

**Fig 9. Fagan nomogram to determine the post-test probabilities of presence of dental caries based on the likelihood ratios.** The post-test probability of a patient having dental caries was 79% with the positive test result and post-test probability of a patient having dental caries was 6% with the negative test result.

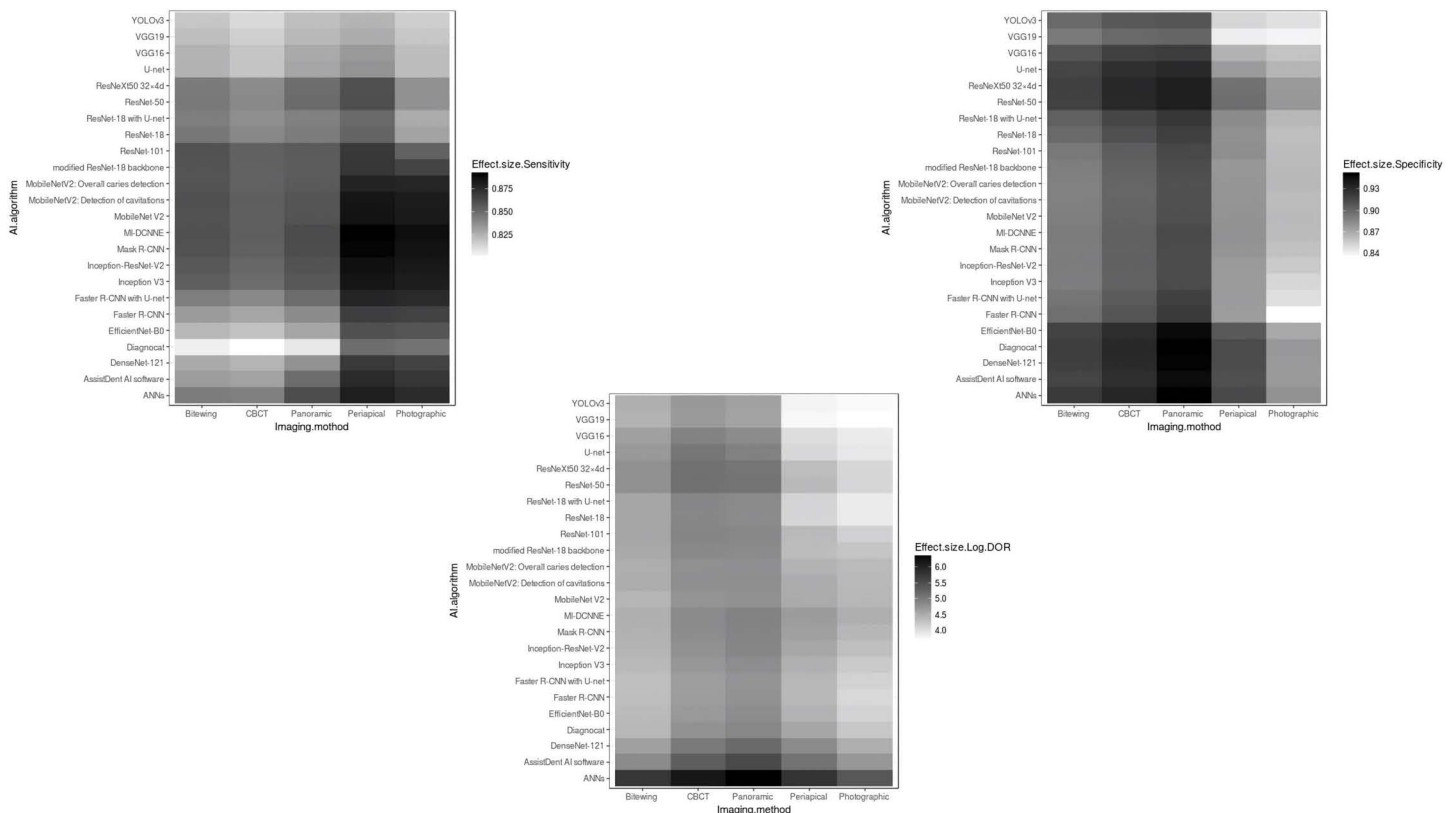

**Fig 10. Pinned scatter plot of the bivariate partial prediction of interaction between imaging method and AI algorithm for sensitivity, specificity, and log diagnostic odds ratio (DOR).** This plot was created by means of random forests model (a machine learning algorithm) (Number of trees in forest: 500, Minimum terminal node size: 5).

Finally, a recent survey published in Nature involving 1,600 researchers from around the world concluded that scientists are worried and expressed fears over the lack of transparency and the 'Black Box Effect' among AI systems, over training data including biased information, AI spreading misinformation, and AI-generated deepfakes [77]. An editorial in the journal Science stated that "*Excitement about AI has been tempered by concerns about potential downsides*" [78]. The dental research community, journal editors, clinicians, and policymakers should be aware and be attentive of the significant and emerging concerns regarding AI safety [79] and ethical worries regarding the use of AI systems in biomedical research and clinical practice [80,81]. Although AI can assist in dental caries diagnosis, it should not be a substitute for human judgment and dental practitioners must take responsibility for the use of AI in caries diagnosis.

## Conclusions

In this umbrella meta-analysis, the analysis of 29423 diagnostic tests resulted in a pooled sensitivity of 0.85, specificity of 0.90, log diagnostic odd ratio of 4.37, AUC of 0.86, positive post-test probability of 79%, and negative post-test probability

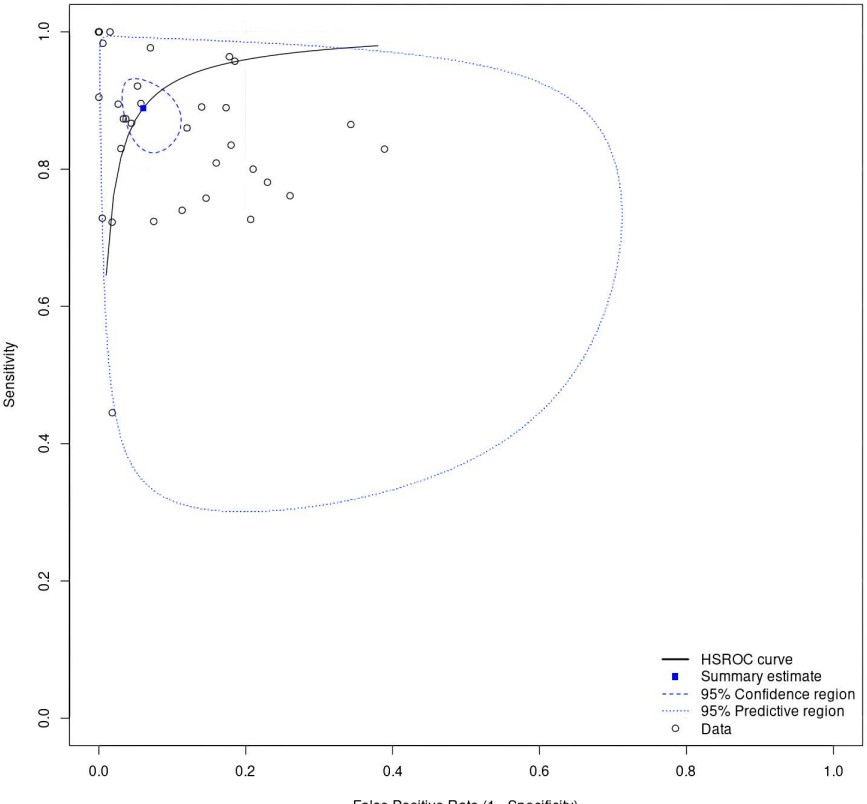

**Fig 11. Summary receiver operating characteristic plot showing meta-analysis performance (HSROC: Hierarchical summary receiver operating characteristic).**

of 6%, which support the implementation of this diagnostic tool in clinical practice. Future studies should focus on specific subpopulations, depth of caries, and real-world performance validation. Although AI can assist in dental caries diagnosis, it should not be a substitute for human judgment.

## Supporting information

**S1 Table. The citation matrix of primary studies included in the systematic reviews for the use of AI in the detection of dental caries.** The "1" implies a checkmark, that is the study is included "0" implies that the study is not included in the review in question.
(DOCX)

**S1 Fig. Visualization of the pairwise CCA (%) with a heatmap.**
(TIF)

**S1 File. PRISMA+DTA+Checklist.**
(DOCX)

## Acknowledgments

The authors would like to extend their gratitude to Prof. Roya Kelishadi for her invaluable guidance and support in the preparation of this manuscript.

## Author contributions

**Conceptualization:** Sarah Arzani, Jafar Kolahi.

**Data curation:** Sarah Arzani, Ali Karimi, Maryam Yazdi, Heejung Bang.

**Formal analysis:** Ali Karimi, Pedram Iranmanesh, Maryam Yazdi, Jafar Kolahi, Heejung Bang.

**Investigation:** Sarah Arzani, Ali Karimi, Pedram Iranmanesh, Mike A Sabeti, Mohammad Hossein Nekoofar, Paul MH Dummer.

**Methodology:** Sarah Arzani, Pedram Iranmanesh, Maryam Yazdi, Jafar Kolahi, Heejung Bang, Paul MH Dummer.

**Project administration:** Sarah Arzani, Jafar Kolahi.

**Resources:** Sarah Arzani, Jafar Kolahi.

**Software:** Sarah Arzani, Maryam Yazdi, Jafar Kolahi.

**Supervision:** Pedram Iranmanesh, Mike A Sabeti, Mohammad Hossein Nekoofar, Jafar Kolahi, Paul MH Dummer.

**Validation:** Sarah Arzani, Pedram Iranmanesh, Mike A Sabeti, Mohammad Hossein Nekoofar, Jafar Kolahi, Heejung Bang, Paul MH Dummer.

**Visualization:** Mike A Sabeti, Mohammad Hossein Nekoofar, Jafar Kolahi, Paul MH Dummer.

**Writing – original draft:** Sarah Arzani, Ali Karimi.

**Writing – review & editing:** Sarah Arzani, Pedram Iranmanesh, Maryam Yazdi, Mike A Sabeti, Mohammad Hossein Nekoofar, Jafar Kolahi, Heejung Bang, Paul MH Dummer.

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
