## [Decision Letter · Decision Letter 0]

11 Jun 2025

Dear Dr. Arzani,

Thank you for submitting your manuscript to PLOS ONE. After careful consideration, we feel that it has merit but does not fully meet PLOS ONE’s publication criteria as it currently stands. Therefore, we invite you to submit a revised version of the manuscript that addresses the points raised during the review process.

We look forward to receiving your revised manuscript.

Kind regards,

Giang Truong Vu, D.D.S., M.S., Ph.D.

Academic Editor

PLOS ONE

Journal Requirements:

2. Please remove all personal information, ensure that the data shared are in accordance with participant consent, and re-upload a fully anonymized data set.

Reviewers' comments:

Reviewer's Responses to Questions

**Comments to the Author**

1. Is the manuscript technically sound, and do the data support the conclusions?

Reviewer #1: Yes

Reviewer #2: Yes

2. Has the statistical analysis been performed appropriately and rigorously?

Reviewer #1: Yes

Reviewer #2: Yes

3. Have the authors made all data underlying the findings in their manuscript fully available?

Reviewer #1: Yes

Reviewer #2: Yes

4. Is the manuscript presented in an intelligible fashion and written in standard English?

Reviewer #1: Yes

Reviewer #2: Yes

Reviewer #1: The manuscript is technically sound and well written. However, some minor corrections would improve the quality.

Heterogeneity has been discussed, however the implications of high variability across studies (e.g., imaging modality, AI model type) could be discussed further. The impact of publication bias on the pooled estimates should be elaborated.

The references have to be formatted according to journal guidelines.

Reviewer #2: Review Comments to the Author:

The topic of this manuscript is timely and relevant. The authors present a broad summary of existing evidence on the use of artificial intelligence (AI) for detecting dental caries, which seems important and useful for the field.

Strengths:

The methodology is clearly described, and the tools used for analysis appear to be appropriate and well chosen for the study’s objective.

The conclusions are well linked to the presented data, and the results are carefully analyzed.

It is also appreciated that the underlying data are fully available.

Suggestions:

It might be helpful to discuss more clearly the limitations related to the fact that only a small number of original studies were included in the meta-analysis.

A short explanation on how overlapping studies across the included reviews were handled could also be useful.

The manuscript would benefit from more critical reflection on how the presence or absence of caries was determined in the original studies. There is little detail on the reference standard, and it remains unclear whether definitions of caries were consistent, validated, or clinically reliable.

It would strengthen the discussion to acknowledge that the most reliable conclusions would come from reviews comparing studies conducted under the same diagnostic protocols and caries definitions.

The type of dental imaging strongly influences diagnostic performance. AI evaluating panoramic radiographs (with tooth overlap) operates under very different conditions than AI analyzing bitewing images (with clear interproximal visibility). Pooling such results may reduce the clinical interpretability of the findings and should be acknowledged as a limitation.

In future analyses, the most valuable evidence will likely come from studies that simulate real-world diagnostic conditions, including time pressure, clinician–patient interaction, and the integration of AI as part of the clinical workflow rather than in isolated image review.

A recently published study demonstrates the importance of incorporating such clinical constraints in AI evaluation and could be considered for citation:

PMID: 40095536

Conclusion:

Overall, this seems to be a well-prepared and solid piece of work. I recommend it for publication after minor clarifications.

**Do you want your identity to be public for this peer review?** For information about this choice, including consent withdrawal, please see our Privacy Policy

Reviewer #1: No

Reviewer #2: No

---

## [Author Response · Author response to Decision Letter 1]

4 Jul 2025

The manuscript was revised based on the journal requirements and guidlines.

Comments to the Author

Reviewer #1: The manuscript is technically sound and well written. However, some minor corrections would improve the quality.

Heterogeneity has been discussed, however the implications of high variability across studies (e.g., imaging modality, AI model type) could be discussed further.

Discussion section (page 22):

-High levels of heterogeneity (I2>95) were found regarding sensitivity, specificity, and log diagnostic odds ratio among the included studies. These levels of heterogeneity can be explained by diversity in imaging modalities (e.g., bitewing vs. panoramic radiographs), differences in AI algorithm architectures, and training dataset characteristics.

The impact of publication bias on the pooled estimates should be elaborated.

Results section (page 17):

-The nonparametric trim-and-fill analysis was employed to assess the impact of publication bias on the pooled estimates and results were presented at Table 6

The references have to be formatted according to journal guidelines.

-Done

Reviewer #2: Review Comments to the Author:

The topic of this manuscript is timely and relevant. The authors present a broad summary of existing evidence on the use of artificial intelligence (AI) for detecting dental caries, which seems important and useful for the field.

Strengths:

The methodology is clearly described, and the tools used for analysis appear to be appropriate and well chosen for the study’s objective.

The conclusions are well linked to the presented data, and the results are carefully analyzed.

It is also appreciated that the underlying data are fully available.

Suggestions:

It might be helpful to discuss more clearly the limitations related to the fact that only a small number of original studies were included in the meta-analysis.

Discussion section (page 22):

-The main limitation of this umbrella review and meta-analysis include the fact that only 20 original studies were included in meta-analysis quantitative data synthesis. The number of primary original research studies was 137, with only 20 (14.5 %) original research articles reporting the necessary details of AI–based caries diagnostic test results including numbers TP, TN, FP, and FN involved in the meta-analysis. To facilitate future meta-analysis, authors are encouraged to report details of AI–based caries diagnostic tests including numbers of TP, TN, FP, and FN, all of which are essential for a meta-analysis.

A short explanation on how overlapping studies across the included reviews were handled could also be useful.

Methods section (page 7):

-Overlapping studies across the included reviews were handled by creation of citation matrix and calculation of corrected covered area (CCA) using ccaR package of R software (R Foundation for Statistical Computing, Vienna, Austria).

Results section (page 15):

-The citation matrix of primary studies included in the systematic reviews was presented at Supplementary Table 1. The CCA_Proportion was 0.07 and CCA_Percentage was 6.90, showing moderate overlap. The pairwise CCA presented at Supplementary Figure 1, showed which combinations of paired reviews had the highest overlap.

The manuscript would benefit from more critical reflection on how the presence or absence of caries was determined in the original studies. There is little detail on the reference standard, and it remains unclear whether definitions of caries were consistent, validated, or clinically reliable.

Results section (page 15):

-Table 5 was added to manuscript. At this table the caries detection method, reference standard and AI algorithm were presented.

It would strengthen the discussion to acknowledge that the most reliable conclusions would come from reviews comparing studies conducted under the same diagnostic protocols and caries definitions.

Discussion section (page 20):

-It is well-known that, the most reliable conclusions would come from reviews comparing studies conducted under the same diagnostic protocols and caries definitions. As showed at Table 5, caries detection method, reference standard and AI algorithm were same among included studies in meta-analysis.

The type of dental imaging strongly influences diagnostic performance. AI evaluating panoramic radiographs (with tooth overlap) operates under very different conditions than AI analyzing bitewing images (with clear interproximal visibility). Pooling such results may reduce the clinical interpretability of the findings and should be acknowledged as a limitation.

Discussion section (page 19):

-Readers must note, the type of dental imaging strongly influences diagnostic performance. AI evaluating panoramic radiographs (with tooth overlap) operates under very different conditions than AI analyzing bitewing images (with clear interproximal visibility). Pooling such results may reduce the clinical interpretability of the findings. To address this issue we conduct subgroup meta-analysis for different imaging modalities and results presented at Figure 4.

In future analyses, the most valuable evidence will likely come from studies that simulate real-world diagnostic conditions, including time pressure, clinician–patient interaction, and the integration of AI as part of the clinical workflow rather than in isolated image review.

A recently published study demonstrates the importance of incorporating such clinical constraints in AI evaluation and could be considered for citation:

-Done

---

## [Decision Letter · Decision Letter 1]

25 Jul 2025

Examining the diagnostic accuracy of artificial intelligence for detecting dental caries across a range of imaging modalities: An umbrella review with meta-analysis

PONE-D-25-05467R1

Dear Dr. Arzani,

We’re pleased to inform you that your manuscript has been judged scientifically suitable for publication and will be formally accepted for publication once it meets all outstanding technical requirements.

Kind regards,

Giang Truong Vu, D.D.S., M.S., Ph.D.

Academic Editor

PLOS ONE

---

## [Editor Report · Acceptance letter]

PONE-D-25-05467R1

PLOS ONE

Dear Dr. Arzani,

I'm pleased to inform you that your manuscript has been deemed suitable for publication in PLOS ONE. Congratulations! Your manuscript is now being handed over to our production team.

Kind regards,

on behalf of

Dr. Giang Truong Vu

Academic Editor

PLOS ONE